# Prevalence and severity of secondary traumatic stress and optimism in Indian health care professionals during COVID-19 lockdown

**Manohar K. N.[1], Neha Parashar** [2]*, **C. R. Satish Kumar[2], Vivek Verma** [3,4], **Sanjiv Rao[1], Sekhar Y.[1], Vijay Kumar K.[5], Amalselvam A.[6], Hemkumar T. R.[7], Prem Kumar B. N.[8], Sridhar K.[9], Pradeep Kumar S.[10], Sangeeta K.[1], Shivam[9], Chetan Kumar[11], Judith[1]**

**1** Manipal Hospital, Bengaluru, India, **2** University of Maastricht, Bengaluru, India, **3** Assam University, Silchar, Assam, India, **4** Department of Neurology, All India Institute of Medical Sciences (AIIMS), New Delhi, India, **5** Narayana Medical Centre, Bengaluru, India, **6** Joseph Medical Centre, Bengaluru, India, **7** Sakra World Hospital, Bengaluru, India, **8** Kempegowda Institute of Medical Sciences, Bengaluru, India, **9** Sharada Medical Centre, Bengaluru, India, **10** Fortis Hospital, Bengaluru, India, **11** Vydehi Medical College, Bengaluru, India

* n.parashar@maastrichtuniversity.nl

**Data Availability Statement:** All relevant data are within the paper and its Supporting information files.

## Abstract

### Background

The COVID-19 pandemic has brought to light the lacunae in the preparedness of healthcare systems across the globe. This preparedness also includes the safety of healthcare providers (HCPs) at various levels. Sudden spread of COVID-19 infection has created threatening and vulnerable conditions for the HCPs. The current pandemic situation has not only affected physical health of HCPs but also their mental health.

### Objective

This study aims to understand the prevalence and severity of secondary traumatic stress, optimism parameters, along with states of mood experienced by the HCPs, *viz.*, doctors, nurses and allied healthcare professionals (including Physiotherapist, Lab technicians, Phlebotomist, dieticians, administrative staff and clinical pharmacist), during the COVID-19 lockdown in India.

### Methodology

The assessment of level of secondary traumatic stress (STS), optimism/pessimism (via Life Orientation Test-Revised) and current mood states experienced by Indian HCPs in the present COVID-19 pandemic situation was done using a primary data of 2,008 HCPs from India during the first lockdown during April-May 2020. Data was collected through snow-ball sampling technique, reaching out to various medical health care professionals through social media platforms.

**Funding:** The author(s) received no specific funding for this work.

**Competing interests:** The authors have declared that no competing interests exist.

## Result

Amongst the study sample 88.2% of doctors, 79.2 of nurses and 58.6% of allied HCPs were found to have STS in varying severity. There was a female preponderance in the category of Severe STS. Higher optimism on the LOTR scale was observed among doctors at 39.3% followed by nurses at 26.7% and allied health care professionals 22.8%. The mood visual analogue scale which measures the "mood" during the survey indicated moderate mood states without any gender bias in the study sample.

## Conclusion

The current investigation sheds light on the magnitude of the STSS experienced by the HCPs in the Indian Subcontinent during the pandemic. This hitherto undiagnosed and unaddressed issue, calls for a dire need of creating better and accessible mental health programmes and facilities for the health care providers in India.

## 1. Introduction

The pandemic induced mayhem including the lockdowns poses challenges to all. If the public is worried about "life vs livelihood," it also burdens the health care professionals differently. Rising up to the occasion during these trying times, and also the inherent risk of getting the disease for self and their own close families has its own effect on the psyche of a HCP. The psychological burden and trauma inflicted upon the treating health care professionals gets translated into secondary/vicarious trauma [1, 2].

Recent studies on Chinese health care professionals who dealt with the first and largest outbreak of the COVID-19 infections had shown that the frontline health care workers such as doctors and nurses faced depressive symptoms, insomnia and anxiety as compared to non-frontline healthcare workers [3–5]. The findings are also supported by a meta-analysis and systematic review done by Pappa and colleagues which indicated the prevalence of anxiety, insomnia and depression amongst primary health care professionals [6]. Similar results were also found in a large scale survey done on Indian health care professionals where they found that the health care professionals reported higher rates of burnouts during the COVID era [7, 8].

Secondary Traumatic stress (STS) is a neglected entity experienced by the HCPs during this unprecedented situation. STS is defined as a natural consequent behaviour and emotions resulting from knowledge about a traumatising event experienced by a significant other. It is stress resulting from helping or wanting to help a traumatised or suffering person and it comprises of symptoms including intrusion, avoidance, and arousal [9]. These STS symptoms share similarities with those of posttraumatic stress disorder (*PTSD*), as suggested in the 4th edition of the Diagnostic and Statistical Manual of Mental Disorders [DSM-IV-TR; APA] [10]. However, unlike PTSD, STS could be due to indirect contact in a professional context (e.g., caring for a traumatized patient) [11].

While dealing with the COVID-19 affected patients; the roles of HCPs evolved drastically, now venturing into areas other than their area of specialization due to the lack of professionals available to keep up with the demand. Besides uncertainties about the disease, its outcomes and more importantly the knowledge of risking self and their families creates occupational stress among HCPs [12–16]. Reports suggest that in addition to work stress, HCPs has faced

new problems such as being verbally and physically assaulted by mobs [17]. A shortage in personal protective equipment (PPE) was also is a significant concern among healthcare professionals [18]. Furthermore, improper training and guidelines regarding PPE raised several concerns [19].

One of the major protective factors among healthcare professionals is their optimistic attitude to cope with the stress related to their profession. ***Optimism*** can be defined as the ability to look at the brighter side of things. Specific skills such as optimism, interpersonal skills, hope, and faith can protect one's mental health.

Lack of literature on exploring the mental health status and possible coping factors of the Indian health care professionals during the pandemic and our own personal experiences led to the conceptualisation of the current study. Hence, the present study aims to report secondary traumatic stress levels, optimism, and mood states experienced by the HCPs within the Indian subcontinent during the COVID-19 pandemic.

## 2. Material and method

### 2.1 Participants and procedures

For the present cross sectional study, information associated with the level of secondary traumatic stress (STS), optimism and mood state during COVID-19 among health care professionals in India were collected. Due to lockdown and to reduce human contact and transmission risk related with the disease; online platforms such as Google forms and social media were utilized as a mode of data collection. The Google forms were circulated on various groups and social media (LinkedIn, WhatsApp) and invited health care professionals from different cities across India to complete the questionnaire voluntarily. Additionally, we also sent the questionnaire to many health care professionals who had cooperated with us, and used their contact network to spread the questionnaire, utilizing snowballing method.

For the safeguarding of data (also mentioned in ethical clearance document); all the data was collected from primary research supervisor's institutional email address and every two weeks the data was removed and secured in an external hard drive which was not connected with internet and was not accessible to anyone but the primary researchers.

The respondents were English proficient health care professionals above 20 and below 65 years of age in India (including doctors, nurses, physiotherapist, lab technicians, dieticians, administrative staff and clinical pharmacist). Before collecting responses, in the consent form and safeguard process for maintaining the anonymity of the data; we stated the purpose of the investigation, and responses were collected only after obtaining the consent. This questionnaire was anonymous. The data collection took place in the months of April & May 2020, which was the initial national lockdown, for first wave of COVID-19 to hit India. The survey began on April 16, 2020, and ended on May 15, 2020, when India was in a complete lockdown period due to outbreak of COVID-19. Due to the nature of Google forms any incomplete questionnaires were not accepted. We collected a total of 2153 questionnaire out of which only 2008 were valid and finally used for analysis. We excluded those observations which were inconsistent or were inappropriately filled and those which were not consented.

### 2.2 Ethics statement

The approval was obtained from The Ethics committee of Manipal hospitals, Bangalore (ECR/34/Inst/KA/2013/RR-19). Our investigation process remained anonymous, and no identifiers (such as name, address, email id, phone numbers, name of hospital employed) were collected.

Every participant was informed about and understood the purpose of our investigation before entering the study.

## 3. Measurements

In order to understand the socio-demographic profile of the population, information on individual's age, gender, marital status, occupation (doctors, nurses, and allied healthcare professionals), years of experience, type of practice (clinic and hospital) and their current state of practice were collected. During the COVID-19 lockdown period, the present study has been conducted during lockdown; therefore, it is assumed that all the doctors irrespective of their specialty where engaged in same duties. The mental health status was assessed using below describes scales:

### 3.1 Secondary Traumatic Stress Scale

The Secondary Traumatic Stress Scale (STSS) is a self-report inventory designed to assess the frequency of STS symptoms in professional caregivers. The STSS is a 17-item measure explicitly designed to assess the effects of healthcare providers' exposure to secondary trauma from patient experiences [11a]. Unlike other measures that include items related to burnout or compassion satisfaction, the 17 STSS items correspond to the 17 DSM-IV PTSD symptoms for Criteria B (Intrusion), C (Avoidance), and D (Arousal; American Psychiatric Association) [10]. Respondents indicate how often they experienced each symptom in the past seven days on a Likert-type scale ranging from 1 ("never") to 5 ("very often"). In place of assessing Criterion A of the diagnostic guidelines for PTSD (trauma exposure): The STSS use "prompts" that suit professional's setting, for instance. In order to fit the emergency room environment, the word "client" was changed to "patient" in all relevant items. By replacing Criterion A (trauma exposure) for DSM-IV-TR PTSD with these prompts, the STSS largely mirrors PTSD from a secondary stressor, which is the definition of STS used in the present study. The fact that the STSS closely mirrors the DSM-IV-TR criteria for PTSD allows it to be validly compared—albeit with some caution—to DSM-IV-TR PTSD [20, 21].

In prior research, the STS showed good psychometric properties. The STSS has acceptable psychometrics as measured by convergent (mean $r = 0.39$) and discriminant (mean absolute $r = 0.07$) validity. The STSS has high overall internal consistency based on Cronbach's alpha values ($\alpha = 0.93$), and acceptable internal consistency for the symptom cluster sub-scales (Intrusion, $\alpha = 0.80$; Avoidance, $\alpha = 0.87$; Arousal, $\alpha = 0.83$). Additionally, its tree-structure model is supported by confirmatory factorial analysis, although the factors are inter-correlated [10].

### 3.2 Life orientation test-revised

Life orientation test revised (LOT-R) is a 10-item scale that measures how optimistic or pessimistic people feel about their future. Respondents use a 5-point rating scale (0 = strongly disagree; 4 = strongly agree) to show how much they agree with 10 statements about positive and negative expectations. These statements include "In uncertain times, I usually expect the best" and "If something can go wrong for me, it will." The internal consistency (Cronbach's alpha) ranged between.74 and.78 [22].

### 3.3 Mood (visual analogue scale)

Mood visual analogue scale (VAS) (0- extremely sad to 10- extremely happy) is a psychometric response scale which is used to measure subjective characteristics or attitudes and have been

used in the past for measuring the multitude of disorders. The Cronbach's alpha values test-retest reliability for mood visual analogue scale (VAMS) ranged between 0.71 and 0.80 [23].

## 4. Statistical analysis

For the exploratory analysis, mean and standard deviations of continuous variables and proportions for categorical variables were used to describe the levels of secondary traumatic stress (STS), optimism/ pessimism, and current mood states in the sample based on profession and gender of the healthcare professionals in the study. Secondary Traumatic Stress Scale, Life Orientation Test Revised and Mood Visual Analogue Scale were used to measure the Secondary Traumatic Stress Symptoms (intrusion, avoidance and arousal), optimism parameters & state of mood respectively. Regression analysis was further used to explore changes in secondary traumatic stress, optimism and mood states. Significance of all statistical tests' were defined as bilateral P<0.01. SAS university edition was used to analyse data in the study.

## 5. Results

### 5.1 General characteristics

The number of participants who participated was 2153, of which complete information was available for 2008 (93%) individuals, which is considered as a population for present study. Among the study sample, 1027(51.15%) were females. Mean age was 35.7(± 11.9) years; females (mean 29.7 [± 8.9] years) were younger than males (mean 41.9 [± 11.5] years). The majority were married (60.2%), percentage of married males (80.2%). Most HCPs were nurses (924, 46%) followed by doctors (611, 30.4%) and the remaining were other allied healthcare professionals. The population is classified in three broad categories based on their clinical roles, viz., doctors, nurses and allied healthcare professional (physiotherapist, dentists, lab technicians, dieticians, administrative staff and clinical pharmacists).

In the population, majority (1109, 55.2%) of the respondents were practising at hospitals having ICU facilities, and among them (738, 66.6%) were females. The remaining respondents (899, 44.8%) were practising at hospitals without having ICU facilities, among them majority were males (610, 67.9%). Mean years of experience in the field of HCPs were 11.0[± 15.8] years, females were less experienced than males (mean 7.1 [± 18.7] years) vs mean 15.1 [± 10.4] years) (Table 1).

### 5.2 Secondary traumatic stress

The key clinical characteristics of the present study are the STSS and optimism (LOT-R) levels of the HCPs. Secondary traumatic stress (STS) was experienced by 1548 (77%) of the HCPs. The doctors and nurses showed more STS than others HCPs, and STS decreased with increase in the age. In the study sample, on STS Categorisation—among doctors, 11.8% had no STS, and 19% had Severe STS, among the Nurses 20.8% had no STS, and 8.2% had severe STS, among the Allied HCPs 41.4% had no STS and only 7.4% had severe STS.

In the study sample, there was a male preponderance for "Mild STS" (above 30%) amongst doctors and allied health care professionals, but among the nurses there was female preponderance (above 40%).

In "moderate STS" category, female preponderance (above 20%) was noted amongst doctors and allied health professional and males showed preponderance in the nursing category (Fig 1).

**Table 1. Mean and standard deviations of the scores obtained on socio-demographic details along with overall secondary traumatic stress (STS) and mood visual analog scale responses.**

| Name of Characteristics | Total | | Female | | Male | | p-value* (based on $\chi^2$ / t-test) |
|---|---|---|---|---|---|---|---|
| | N | % | N | % | N | % | |
| Age mean (SD) | 35.7 (11.9) | | 29.7 (8.9) | | 41.9 (11.5) | | |
| Marital status | | | | | | | |
| • Married | 1208 | 60.2 | 421 | 41.0 | 787 | 80.2 | < .0001 |
| • Unmarried | 800 | 39.8 | 606 | 59.0 | 194 | 19.8 | < .0001 |
| Clinical role | | | | | | | |
| • Doctor | 611 | 30.4 | 198 | 19.3 | 413 | 42.1 | < .0001 |
| • Nurse | 924 | 46.0 | 783 | 76.2 | 141 | 14.4 | < .0001 |
| • Others | 473 | 23.6 | 46 | 4.5 | 427 | 43.5 | < .0001 |
| Type of practice | | | | | | | |
| • Hospital without ICU | 899 | 44.8 | 289 | 28.1 | 610 | 62.2 | < .0001 |
| • Hospital with ICU | 1109 | 55.2 | 738 | 17.9 | 371 | 37.8 | < .0001 |
| Experience in the field (years) (Mean (SD)) | 11.0 (15.8) | | 7.1 (18.7) | | 15.1 (10.4) | | < .0001 |
| Intrusion (Mean (SD)) | 10.2(3.8) | | 10.7(3.3) | | 9.6(4.1) | | < .0001 |
| Avoidance (Mean (SD)) | 14.4(4.6) | | 14.6(4.3) | | 14.2(4.9) | | < .0001 |
| Arousal (Mean (SD)) | 10.8(3.6) | | 10.7(3.4) | | 10.8(3.7) | | < .0001 |
| Mood VAS (Mean (SD)) | 5.7(2.3) | | 5.7(2.3) | | 5.7(2.2) | | < .0001 |

*(Chi square test was used to analyse categorical data, whereas t-test was used to analyse differences in means between groups).

In "high STS" category there was no gender bias amongst doctors and nurses, whereas female preponderance was noted among the allied health care professionals.

"High" (74, 12.1%) and "Moderate STS" level (129, 21.1%) were also found more among doctors than that of nurses and allied healthcare professionals (Table 2).

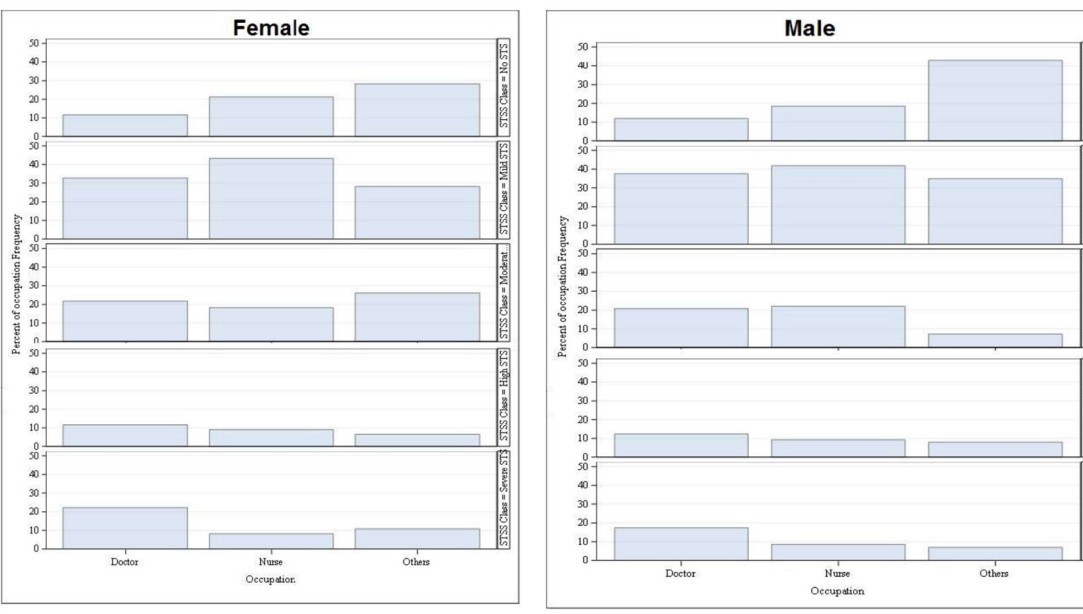

**Fig 1. Occupation-wise distribution of severity of secondary traumatic stress of HCPs based on their gender.**

**Table 2. Descriptive data, including frequency and percentage for the health care professionals and severity of secondary traumatic stress and varying levels of optimism/pessimism using Life Orientation Test-Revised.**

| Variable | Occupation | | | | p-value* (based on $\chi^2$ / t-test) |
|---|---|---|---|---|---|
| | Total (2008) | Doctor (n = 611) | Nurse (n = 924) | Allied HCP (n = 473) | |
| **STSS** | | | | | < .0001 |
| • No STS | 460 | 72 (11.8) | 192 (20.8) | 196 (41.4) | |
| • Mild STS | 780 | 220 (36.0) | 398 (43.1) | 162 (34.3) | |
| • Moderate STS | 346 | 129 (21.1) | 174 (18.8) | 43 (9.1) | |
| • High STS | 195 | 74 (12.1) | 84 (9.1) | 37 (7.8) | |
| • Severe STS | 227 | 116 (19.0) | 76 (8.2) | 35 (7.4) | |
| **Intrusion (mean(SD))** | | 11.7(3.7) | 10.4(3.1) | 7.8(3.9) | < .0001 |
| **Avoidance(mean(SD))** | | 15.8(4.8) | 14.4(4.0) | 12.7(4.8) | < .0001 |
| **Arousal(mean(SD))** | | 11.8(3.7) | 10.3(3.2) | 10.2(3.8) | < .0001 |
| **LOTR** | | | | | < .0001 |
| • Low Optimism (High pessimism) | 226 | 67 (11.0) | 86 (9.3) | 73 (15.4) | |
| • Moderate Optimism | 1183 | 300 (49.1) | 591 (64.0) | 292 (61.7) | |
| • High Optimism (Low pessimism) | 599 | 244 (39.3) | 247 (26.7) | 108 (22.8) | |
| **Mood VAS** | | 5.8(2.2) | 5.9(2.4) | 5.2(2.0) | < .0001 |

*(Chi square test was used to analyse categorical data, whereas t-test was used to analyse differences in means between groups).

The doctors were found to be high on "Severe STS" level (116, 19.0%), followed by nurses (76, 8.2%) and the other allied healthcare professionals (35, 7.4%). In "severe STS" female doctors and female allied health care professionals were noted to be higher than their male counterparts (10%).

Mean score on Intrusion Scale, which measures intrusive thoughts related to trauma, flashbacks and recollections was found to be 10.2 (±3.75); of which female reported a mean score of 10.7 (± 3.33), which is slightly higher than those of males (mean score of 9.62 (± 4.08)). This indicated high intrusive thoughts among females as compared to males.

On the Avoidance Scale, which measures the attempts to avoid any stimuli or triggers that might be related to the traumatic event, the participants reported an overall mean score of 14.4 (±4.6); mean score of females being 14.6 (±4.3) whereas males mean score being 14.2 (±4.9); indicating both males and female utilizing avoidance as a coping strategy.

On the arousal scale which indicates jumpiness, irritability, insomnia, decreased concentration and hyper vigilance the participants reported a mean score of 10.8 (± 3.56); females reporting a mean score of 10.7(± 3.4) and males mean score of 10.8(± 3.7) (Table 1).

## 5.3 Optimism (LOT-R) and perceived mood state

Life Orientation Test-Revised (LOT-R) is another key component, which measures the dispositional optimism of an individual. Table 3 shows that among the HCPs, high Optimism was mostly observed among doctors (244, 39.3%), followed by nurses (247, 26.7%) and allied healthcare professionals (108, 22.8%), whereas, in cases of low optimism category, the order changed i.e., allied healthcare professionals (73, 15.4%), followed by doctors (67, 11.0%) and nurses (86, 9.3%) (Fig 2).

The perceived mood state of the HCPs was assessed with the help of a mood visual analogue scale (11-point Likert scale; where 0 = extremely sad and 11 = extremely happy); the overall mean for the sample was found to be 5.68 (± 2.26) indicating moderate mood states reported

Table 3. Describing the categories of secondary traumatic stress levels based on cut off scores of the secondary traumatic stress scale [11b].

| Category of STS | STSS Score |
| --- | --- |
| 1. Little or No | < 28 |
| 2. Mild | 28 to 37 |
| 3. Moderate | 38 to 43 |
| 4. High | 44 to 48 |
| 5. Severe | > 49 |

by participants at the time of taking the survey. Gender-wise mean mood VAS score were similar between the groups [females: 5.71 (± 2.34) and males: 5.65 (± 2.18)].

In mood VAS among females, more nurses were having "neutral" mood than the doctors and allied HCPs. On the analysis of happiness based on the VAS score, we found nurses to be happier than doctors and allied HCPs. In mood VAS, more allied male HCPs had "neutral" mood as compared to doctors and nurses (Fig 3).

To summarise the regression analysis, doctors and nurses showed happier mood when compared to others HCPs. In STS, doctors and nurses showed more STS than others HCPs, and STS decreased with increase in the age. The doctors and nurses had shown higher optimism than others HCPs. Females HCPs experienced higher "sad" mood as compared to males (Table 4).

## 6. Discussion

COVID-19 came with a threat package of being highly contagious in nature with rapid spread across the globe, and warranted an unprecedented situation to be faced by the medical fraternity. The current study showed that 77% (N = 1548) of HCPs (doctors, nurses, and allied health care professional) reported prevalence of STS. Severe STS was reported at a higher rate

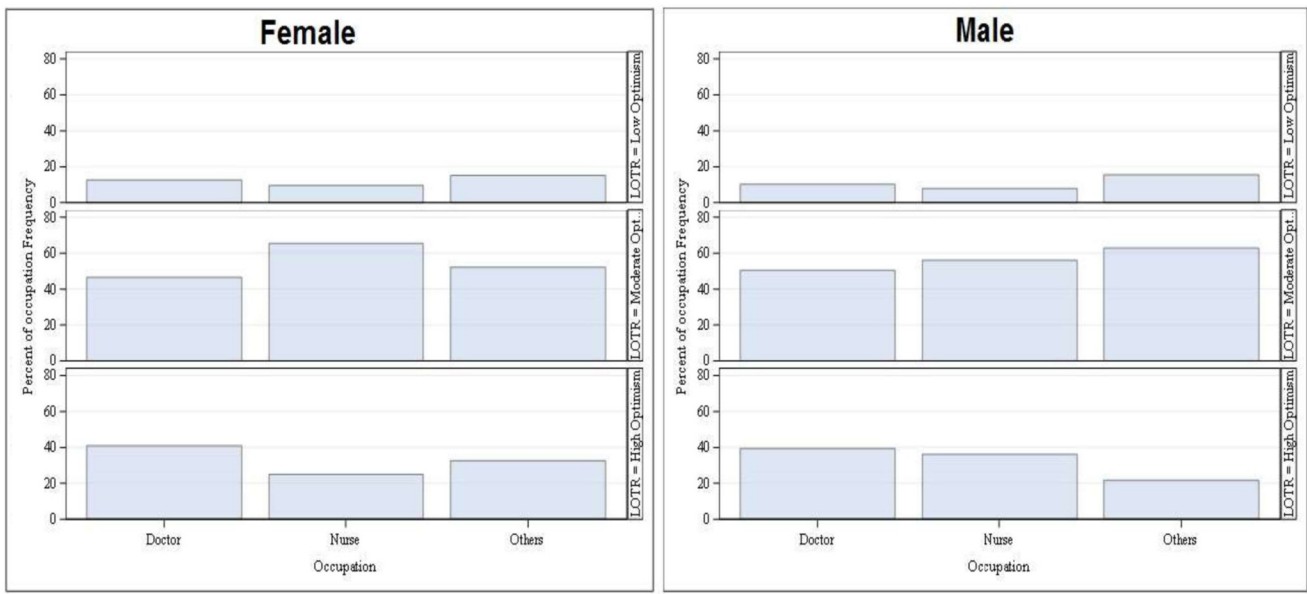

Fig 2. Occupation-wise distribution of varying levels of optimism/pessimism using Life Orientation Test-Revised of HCPs based on their gender.

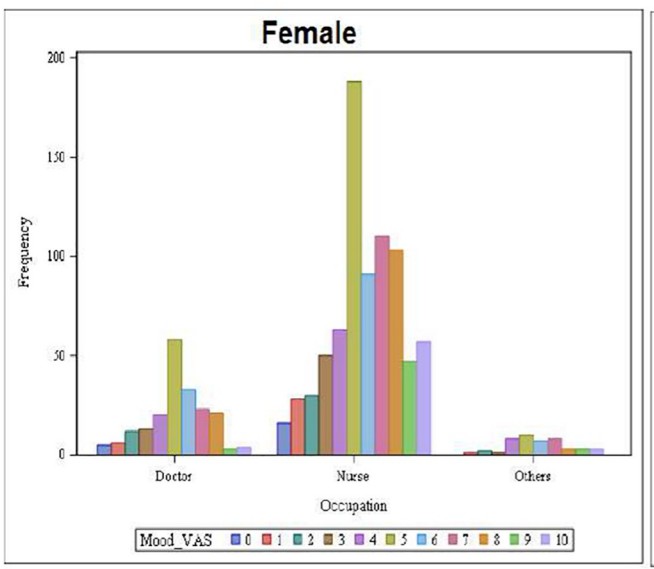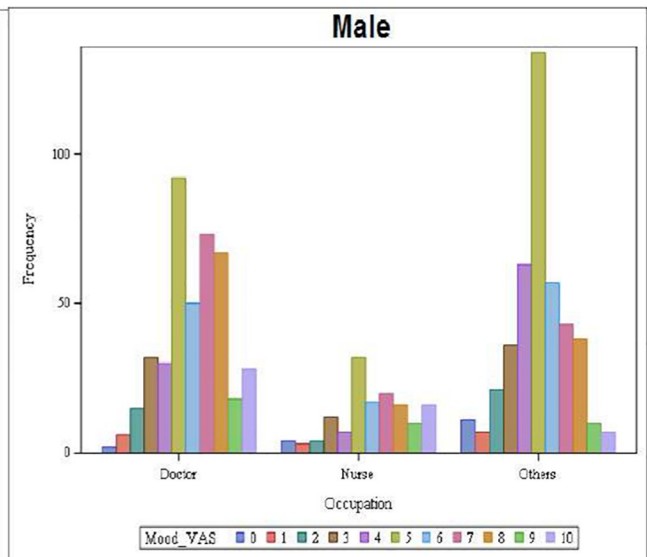

**Fig 3. Occupation-wise distribution of mood status of HCPs based on their gender.**

among doctors as compared to nurses and allied HCPs, which is similar to the earlier study findings published during the pandemic [24].

There was a difference in the patterns of responses among female and male participants who showed signs of intrusive thoughts, using avoidance as a coping mechanism and arousal

**Table 4. Details of regression model for STSS, LOTR and mood VAS and its associated covariates selected through stepwise procedure.**

| Parameter | | Estimates | Standard Error | P value |
|---|---|---|---|---|
| **STSS** | | | | |
| **Intercept** | | 35.74231 | 1.134083 | < .0001 |
| **Age** | | -0.1168 | 0.023984 | < .0001 |
| **Occupation** | | | | |
| | Doctor | 8.496866 | 0.611651 | < .0001 |
| | Nurse | 2.556519 | 0.678363 | 0.0002 |
| | Others[R] | | | |
| **LOTR** | | | | |
| **Intercept** | | 2.073996 | 0.027992 | < .0001 |
| **Occupation** | | | | |
| | Doctor | 0.215693 | 0.037284 | < .0001 |
| | Nurse | 0.100247 | 0.034419 | 0.0036 |
| | Others[R] | | | |
| **Mood_VAS** | | | | |
| **Intercept** | | 5.226624 | 0.103963 | < .0001 |
| **Gender** | | | | |
| | Female | -0.37377 | 0.13069 | 0.0043 |
| | Male[R] | | | |
| **Occupation** | | | | |
| | Doctor | 0.678459 | 0.140597 | < .0001 |
| | Nurse | 0.953744 | 0.160338 | < .0001 |
| | Others[R] | | | |

when faced by triggers in the environment. The results show that female health care professionals showed higher levels of secondary traumatic stress (also related to symptoms of post-traumatic stress) as compared to their male counterparts, especially doctors and nurses as compared to other health care professionals. In the other studies conducted in India related to burnout and distress among doctors and nurses during the time of COVID-19, the health care professionals also showed significant burnout due to their direct contact and involvement in their work with pandemic related work and patient involvement [7, 8]. Though no comparative data exist for pre-COVID-19 or pandemic and related secondary traumatization studies in Indian health care professionals, earlier studies suggest that burnout is associated with professional life experienced by doctors and nurses in India [25–27].

Various Indian researches including a systematic review and meta-analysis (during COVID-19) also throws light on the presence of high levels of stress-related disorder among health care workers such as anxiety, depression, insomnia, hopelessness during the pandemic [28–30].

A study by Li et al. informed that vicarious traumatization (STS) adversely affected both medical and non-medical staff; also the vicarious traumatization was worse in non-front line medical workers as compared by frontline medical staff [31].

The current study sheds light on the reported mood states along with the traumatic stress and pessimism, experienced during patient care by the HCPs. The results show that neutral moods were recorded across spectrum between both male and female health care professionals. The findings are in line with the study among healthcare professionals during the pandemic conditions in India; where they showed signs of various mood and anxiety disorder like symptoms [24, 32].

The STS and burnouts have been reported higher in other studies as well, where the data collection was during a similar period of COVID-19 spread peak [33]. The results of our study are consistent with the studies done on nursing students during the SARS pandemic [34].

## 6.1 Suggested intervention

As the pandemic peaks, the disease related psychological burden also spirals high in the neglected healthcare providers. Age related variance and marital status contributes to fear of transmission of the disease to the family and job insecurity [35]. Identifying and addressing these mental health issues and ensuring both physical and psychological safety should become the priority for not only front liners but for everyone in the field of medicine.

The deleterious effect of the pandemic on the mental health status of HCPs is important to be addressed on a war footing. Early recognition and intervention to tackle these issues would go a long way to prepare the HCPs to cope with this situation and to give their best to the society. High level of optimism helps to cope with pandemic stress and foster lower level of psychological problems [36]. Resources such as psycho-social support, leisure time and improvement in infrastructure adaptations in hospitals could help improve their mental health.

No war could be won if warriors are fighting demons within themselves. We propose the following interventions:

1. Early Recognition of the Mental health issues of HCPs and their families.

2. Easy, Free and Confidential access to the counsellors / psychologists/ psychiatrists.

3. Building a peer network within the co-workers to provide a psychological support system at work place.

4. Emphasise on Work—Life Balance.

5. Positive Reinforcement System.

## 7. Limitations

Irrespective of the large data set and strength of the study, there are certain limitations. Firstly, utilization of the cross-sectional design, lack of homogeneity at various levels, and over-representation of a particular group of healthcare providers, could have played a mediating role in the results, interfering with causality analysis between the variables of the study.

Secondly, the participants recruited with the help of Google forms shared on social media with snowballing effect, by virtue of this methodology utilised, an over representation of technology savvy participants could have happened contributing to the bias. We have not excluded patients with prior psychiatric ailments and addictions.

Besides, psychological health is influenced by various factors, including the personal and professional situation, besides the situation created by the pandemic, it has increased the workload and safety concerns, other factors such as family support, job stress, disturbed daily activities could have contributed significantly to the overall health and quality of life.

## 8. Conclusion

This study sheds light on the levels of distress and secondary trauma experienced by healthcare professionals in India during COVID-19 pandemic. Various factors such as the sudden outbreak of the disease, rampant spread, lack of preparedness, uncertain management guidelines, besides risks for self and family, could have been critical intervening factors to create distress and burnout amongst HCPs, making them vulnerable to various mental health and physical health issues during the pandemic.

There is immediate need for focus on the secondary traumatic stress experienced by the healthcare provider. This study further emphasises the need for social and administration level support in helping to build better healthcare policies to cater the need of HCPs.

This study is a call for Saving the Saviour and a humble request to throw light on the darker side of being HCPs in this current situation. Under current circumstances, it is important to evaluate, understand and prioritize the mental health needs of the health care professionals. Current research highlights the mental health needs of HCPs and calls for the policy makers and administrators to prioritise the mental health interventions for the HCPs, enabling them to not only cope up but also to serve the community at large. This becomes an important study throwing light on the prevailing unaddressed mental health state of HCPs [37, 38].

## Supporting information

**S1 Dataset.**
(XLSX)

## Acknowledgments

The authors would like to thank both of the anonymous reviewers and editorial team for their valuable comments and suggestions that helped us to improve the article's quality.

## Author Contributions

**Conceptualization:** Manohar K. N., Neha Parashar.

**Data curation:** Manohar K. N., Neha Parashar, Vivek Verma, Sanjiv Rao, Sekhar Y., Vijay Kumar K., Amalselvam A., Hemkumar T. R., Prem Kumar B. N., Sridhar K., Pradeep Kumar S., Sangeeta K., Shivam, Chetan Kumar, Judith.

**Formal analysis:** Manohar K. N., C. R. Satish Kumar, Vivek Verma.

**Investigation:** Manohar K. N., Neha Parashar, Vivek Verma.

**Methodology:** Manohar K. N., Neha Parashar.

**Project administration:** Manohar K. N.

**Resources:** Manohar K. N., C. R. Satish Kumar, Sekhar Y., Vijay Kumar K., Amalselvam A.

**Software:** Vivek Verma.

**Supervision:** Manohar K. N.

**Validation:** Vivek Verma.

**Visualization:** Vivek Verma.

**Writing – original draft:** Neha Parashar.

**Writing – review & editing:** Manohar K. N., Neha Parashar, C. R. Satish Kumar, Vivek Verma.

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
