## [Decision Letter · Decision Letter 0]

27 May 2021

PONE-D-21-14477

Prevalence and severity of secondary traumatic stress and optimism in healthcare professionals in India during COVID-19 lockdown.

PLOS ONE

Dear Dr. Parashar,

Thank you for submitting your manuscript to PLOS ONE. After careful consideration, we feel that it has merit but does not fully meet PLOS ONE’s publication criteria as it currently stands. Therefore, we invite you to submit a revised version of the manuscript that addresses the points raised during the review process.

We look forward to receiving your revised manuscript.

Kind regards,

Leeberk Raja Inbaraj, MD

Academic Editor

PLOS ONE

Journal Requirements:

3)  We note that you have indicated that data from this study are available upon request. PLOS only allows data to be available upon request if there are legal or ethical restrictions on sharing data publicly. For more information on unacceptable data access restrictions, please see http://journals.plos.org/plosone/s/data-availability#loc-unacceptable-data-access-restrictions.

4) PLOS requires an ORCID iD for the corresponding author in Editorial Manager on papers submitted after December 6th, 2016. Please ensure that you have an ORCID iD and that it is validated in Editorial Manager. To do this, go to ‘Update my Information’ (in the upper left-hand corner of the main menu), and click on the Fetch/Validate link next to the ORCID field. This will take you to the ORCID site and allow you to create a new iD or authenticate a pre-existing iD in Editorial Manager. Please see the following video for instructions on linking an ORCID iD to your Editorial Manager account: https://www.youtube.com/watch?v=_xcclfuvtxQ

5) Please include your tables as part of your main manuscript and remove the individual files. Please note that supplementary tables (should remain/ be uploaded) as separate "supporting information" files

6) Please amend either the title on the online submission form (via Edit Submission) or the title on the first page of your manuscript (title page) so that they are identical.

Reviewers' comments:

Reviewer's Responses to Questions

**Comments to the Author**

1. Is the manuscript technically sound, and do the data support the conclusions?

Reviewer #1: Yes

Reviewer #2: Yes

2. Has the statistical analysis been performed appropriately and rigorously? 

Reviewer #1: Yes

Reviewer #2: Yes

3. Have the authors made all data underlying the findings in their manuscript fully available?

Reviewer #1: No

Reviewer #2: Yes

4. Is the manuscript presented in an intelligible fashion and written in standard English?

Reviewer #1: Yes

Reviewer #2: Yes

5. Review Comments to the Author

Reviewer #1: Since Doctors and Nurses reported to have high level of stress, more among doctors-author can a make statement about the relationship of specialty of a doctor and experienced stress.

Author can share recommendations about management of stress for HCPs,it will be helpful for policy makers to implement.

Author can specify the Cronbach alpha of the used tests for the study.

Reviewer #2: Authors have gone for a very novel idea which is very much relevant in present times.

There are definitely few grammatical errors which must be addressed.

Authors need to mention more Indian studies - few are from Indian Journal of Psychiatry and Asian Journal of Psychiatry.

6. PLOS authors have the option to publish the peer review history of their article (what does this mean?). If published, this will include your full peer review and any attached files.

Reviewer #1: No

Reviewer #2: **Yes: **Fazle Roub Bhat

---

## [Author Response · Author response to Decision Letter 0]

1 Aug 2021

RESPONSE TO REVIEWERS’REPORT(S)

Reviewer 1 Response

(a) Since Doctors and Nurses reported to have high level of stress, more among doctors-author can a make statement about the relationship of specialty of a doctor and experienced stress. During the COVID-19 lockdown period, all the doctors irrespective of their specialization were involved in COVID duties only. Based on their role in broader perspective the clinical group is classified into three subgroups.

 [Page-5, paragraph 2; line 5-8]

(b) Author can share recommendations about management of stress for HCPs, it will be helpful for policy makers to implement. Page-13, paragraph 4; line 5-10

(c) Author can specify the Cronbach alpha of the used tests for the study. 1. Secondary traumatic Stress Scale (Page-6, paragraph 2)

2. Life Orientation Test-Revised (Page-6, paragraph 3)

3. Mood (VAS) (Page-7, paragraph 1)

New reference added for Mood (VAS) Cronbach score.

23. Flynn D, van Schaik P, van Wersch A. A comparison of multi-item likert and visual analogue scales for the assessment of transactionally defined coping. Eur J Psychol Assess. 2004;20:49–58.

Reviewer 2 Response

There are definitely few grammatical errors which must be addressed. Needful changes have been incorporated at various places in the revised manuscript. (yellow highlights, apart from the changes mentioned in the list).

Authors need to mention more Indian studies - few are from Indian Journal of Psychiatry and Asian Journal of Psychiatry.

 Page-12, paragraph 2 and Page-14, paragraph 4; line 4-10 (multiple new and relevant Indian studies added).

33. Arslan, G., Yıldırım, M., Tanhan, A. et al. Coronavirus Stress, Optimism-Pessimism, Psychological Inflexibility, and Psychological Health: Psychometric Properties of the Coronavirus Stress Measure. Int J Ment Health Addiction .

34. Mathur S, Sharma D, Solanki RK, Goyal MK. Stress-related disorders in health-care workers in COVID-19 pandemic: A cross-sectional study from India. Indian J Med Spec 2020;11:180-4

35. Chatterjee SS, Chakrabarty M, Banerjee D, Grover S, Chatterjee SS, Dan U. Stress, Sleep and Psychological Impact in Healthcare Workers During the Early Phase of COVID-19 in India: A Factor Analysis. Front Psychol. 2021 Feb 25;12:611314.

36. Singh RK, Bajpai R, Kaswan P. COVID-19 pandemic and psychological wellbeing among health care workers and general population: A systematic-review and meta-analysis of the current evidence from India. Clin Epidemiol Glob Health. 2021 Jul-Sep;11:100737.

37. Grover S, Dua D, Shouan A, Nehra R, Avasthi A. Perceived stress and barriers to seeking help from mental health professionals among trainee doctors at a tertiary care centre in North India. Asian J Psychiatr 2019;39:143-9.

38. Banerjee D, Vijayakumar HG, Rao T S. ”Watching the watchmen:” Mental health needs and solutions for the health-care workers during the coronavirus disease 2019 pandemic. Int J Health Allied Sci 2020;9, Suppl S1:51-4

Other comments/ suggestions: 

1. Change in the name from Diagnostic and Statistical Manual of Psychiatric Disorders and DSM-IV”.

2. Mention the timeline of first national lockdown. 1. Page-3, paragraph 43 lines 6 to “Diagnostic and Statistical Manual of Mental Disorders and DSM-IV-TR”.)

2. Page-5, paragraph 1, line 8 (The first lockdown was initially stated to be from 25 March 2020 – 14 April 2020 but it was extended post April 14.-in certain states.)

1.Change in table numbering (as table number 1 related to STSS score was added).

 1. Table 1 : Page 6 (new addition)

2. Earlier Table 1= Current Table 2 (page-8)

3. Earlier Table 2= Current Table 3 (page-10)

4. Earlier Table 3= Current Table 4 (page-11)

*Table numbers were also updated respectively in the revised manuscript (wherever mentioned).

Note: All of the changes in the revised manuscript are highlighted using YELLOWcolour.

---

## [Decision Letter · Decision Letter 1]

1 Sep 2021

Prevalence and Severity of Secondary Traumatic Stress and Optimism in Indian health care professionals during COVID-19 lockdown

PONE-D-21-14477R1

Dear Dr. Parashar,

We’re pleased to inform you that your manuscript has been judged scientifically suitable for publication and will be formally accepted for publication once it meets all outstanding technical requirements.

Kind regards,

Leeberk Raja Inbaraj, MD

Academic Editor

PLOS ONE

Additional Editor Comments (optional):

Reviewers' comments:

Reviewer's Responses to Questions

**Comments to the Author**

1. If the authors have adequately addressed your comments raised in a previous round of review and you feel that this manuscript is now acceptable for publication, you may indicate that here to bypass the “Comments to the Author” section, enter your conflict of interest statement in the “Confidential to Editor” section, and submit your "Accept" recommendation.

Reviewer #1: All comments have been addressed

2. Is the manuscript technically sound, and do the data support the conclusions?

Reviewer #1: Yes

3. Has the statistical analysis been performed appropriately and rigorously? 

Reviewer #1: Yes

4. Have the authors made all data underlying the findings in their manuscript fully available?

Reviewer #1: Yes

5. Is the manuscript presented in an intelligible fashion and written in standard English?

Reviewer #1: Yes

6. Review Comments to the Author

Reviewer #1: Recommended in terms of good statistical analysis, data support the conclusions,the manuscript is written well.

7. PLOS authors have the option to publish the peer review history of their article (what does this mean?). If published, this will include your full peer review and any attached files.

Reviewer #1: No

---

## [Editor Report · Acceptance letter]

16 Sep 2021

PONE-D-21-14477R1 

Prevalence and Severity of Secondary Traumatic Stress and Optimism in Indian health care professionals during COVID-19 lockdown 

Dear Dr. Parashar:

I'm pleased to inform you that your manuscript has been deemed suitable for publication in PLOS ONE. Congratulations! Your manuscript is now with our production department. 

Kind regards, 

on behalf of

Dr. Leeberk Raja Inbaraj 

Academic Editor

PLOS ONE